# Decomposition of Formic Acid and Acetic Acid into Hydrogen Using Graphitic Carbon Nitride Supported Single Metal Catalyst

**Wan Nor Roslam Wan Isahak** [1,*], **Muhammad Nizam Kamaruddin** [1], **Zatil Amali Che Ramli** [2], **Khairul Naim Ahmad** [1], **Waleed Khalid Al-Azzawi** [3] **and Ahmed Al-Amiery** [1,4]

1　Department of Chemical and Process Engineering, Faculty of Engineering and Built Environment, Universiti Kebangsaan Malaysia, Bangi 43600, Selangor, Malaysia
2　Fuel Cell Institute, Universiti Kebangsaan Malaysia, Bangi 43600, Selangor, Malaysia
3　Department of Medical Instruments Engineering Techniques, Al-Farahidi University, Baghdad 10001, Iraq
4　Energy and Renewable Energies Technology Center, University of Technology-Iraq, Baghdad 10001, Iraq
*　Correspondence: wannorroslam@ukm.edu.my

**Abstract:** In a combination of generation and storage of hydrogen gas, both formic acid (FA) and acetic acid (AA) have been notified as efficient hydrogen carriers. This study was conducted to synthesize the monometallic catalysts namely palladium (Pd), copper (Cu), and zinc (Zn) on graphitic-carbon nitride (g-$C_3N_4$) and to study the potential of these catalysts in FA and mixed formic acid (FA)-acetic acid (AA) decomposition reaction. Several parameters have been studied in this work such as the type of active metals, temperature, and metal loadings. The mass percentage of Pd, Cu, and Zn metal used in this experiment are 1, 3, and 5 wt%, respectively. At low temperature of 30 °C, 5 wt% Pd/g-$C_3N_4$ catalyst yielded higher volume of gas with 3.3 mL, instead of other Pd percentage loadings. However, at higher temperature of 70 °C and 98% FA concentration, Pd with 1 wt%, 3 wt%, and 5 wt% of loading over g-$C_3N_4$ has successfully produced optimum gas ($H_2$ and $CO_2$) of 4.3 mL, 7.4 mL, and 4.5 mL in each reaction, respectively. At higher temperature, Pd metal showed high catalytic performance and the most active element of monometallic system in ambient condition. Meanwhile, at higher percentage of Pd metal, the catalytic decomposition reaction also increased thus producing more gas. However, it can be seen the agglomeration of the particles formed at higher loadings of Pd (5 wt%), and remarkably lowering the catalytic activity at higher temperature, while higher activity at low temperature of 30 °C. The result also showed low catalytic decomposition reaction for Cu and Zn catalyst, due to the small formation of Cu and Zn metal, but presence of high metal oxide (CuO) and (ZnO) promotes the passive layer formation on the catalyst surface.

**Keywords:** organic waste; hydrogen carrier; monometallic catalyst; graphitic-carbon nitride; clean process

## 1. Introduction

Global warming continues to gain attention from scientists around the world. Released of toxic and greenhouse gases such as sulphur dioxide ($SO_2$) as well as carbon dioxide ($CO_2$) are the main reason for global warming other than environmental pollution which caused by used of large fossil fuels. An important transformation has been made by scientists around the world to study alternative ways that are better and more effective in reducing the use of fossil fuels by using cleaner fuel sources such as hydrogen [1,2].

Hydrogen is one of the common elements and contributes 75% of the total mass of the universe. Hydrogen is not a source of energy or fuel, but it had been regarded as an energy carrier where it stores energy that had created elsewhere. For example, it reacts with oxygen in the air during combustion and produce heat that will be use as energy. There are various ways to produce hydrogen fuel such as electrolysis using hydroelectric,

solar energy, wind energy, and nuclear energy. The essential benefits of hydrogen fuel used are carbon-free, non-toxic, lighter than air, and easily produced from a variety of different sources [3].

In a combination of generation and storage of hydrogen gas, both formic and acetic acid are known as efficient carrier fluid for hydrogen. Formic acid HCOOH is a type of carboxylic acid that contain 4.4 wt% hydrogen which is lower than DOE 2015 target up to 5.5 wt% [4]. Acetic acid is a type of molecule or a simple liquid with chemical formula of $CH_3COOH$ and it contains 6.7 wt% of hydrogen [5]. Formic acid and acetic commonly presence in organic waste from many processes especially biological process through fermentation to produce biogas or bio-hydrogen. High carboxylic acid content contributes to the environmental problem and utilization of these compounds can eliminate the related environmental issues in the future and turns it into clean hydrogen energy considered a significant value added to the process. Based on thermochemistry considerations, it is shown that formic acid can go through two different decomposition routes. However, decomposition into hydrogen and carbon dioxide gas would be more favorable at lower temperature, while at higher temperature dehydration of formic acid would be take place.

$$HCOOH \rightarrow H_2O + CO \qquad \Delta G_{298K} = -14.9 \text{ kJ mol}^{-1}$$

$$HCOOH \rightarrow H_2 + CO_2 \qquad \Delta G_{298K} = -35.0 \text{ kJ mol}^{-1}$$

Formic acid decomposition occurred at relatively slower rate. This kind of reaction needs a presence of catalysts to speed up the reaction by lowering the activation energy. It also makes the process of bonds breaking and formation of new bonds are more effective [1]. The active heterogeneous catalysts comprise of precious metals and transition metallic element such as Pd, Fe, Co, Ni, and Mn where the catalysts were used in decomposition of formic acid [6,7]. Metallic catalyst usually inhibited due to passive layer formation on the surface such as formate compound.

Graphitic carbon nitride (g-$C_3N_4$) is the most stable allotrope carbon nitride. It also has surface properties that can be used for various applications including as catalyst and photocatalyst reaction as it has high thermal and hydrothermal stability that enable it to function in a liquid, gas, and high temperature environment [8–10]. There are many literatures reported the synthesis method, potential, properties, advantageous, and catalytic/photocatalytic activities using graphitic carbon nitride (g-$C_3N_4$) material [11–13]. In photocatalytic reaction, graphitic carbon nitride has been widely used due to its properties such as suitable band structure, tunable bandgap, and low cost [14]. Nevertheless, modification must be done on bulk g-$C_3N_4$ to increase photocatalytic reaction rate. As described by Ji et al. (2020) in photocatalytic reaction, heterostructure with g-$C_3N_4$ can extend the adsorption to visible light region and retarded electron-hole pairs recombination [15]. While, introducing cyano groups to g-$C_3N_4$ nanosheets increase the charge carrier density, reduce the surface charge transfer resistance, and promote charge transfer and separation [16]. Qiu et al. (2022) reported presence of porous structure in g-$C_3N_4$ with many cyanide groups as a metal-free photocatalyst exhibited synergetic redox by photogenerated electron-hole pairs [17]. In addition, thermal calcination method using ammonium sulfate as porogen contributed to the creation of a porous structure in synthesized g-$C_3N_4$. Then, N-deficient ordered mesoporous g-$C_3N_4$ modified with AgPd nanoparticles with high catalytic and reusability was reported by a group of researcher, Wan et al., 2022 [18]. The high catalytic performance of this catalyst was recognized to the unique structure of N-ompg-$C_3N_4$ with higher surface area and abundant surface defects, the strong metal-support interaction between AgPd and N-ompg-$C_3N_4$, and charge transfer from Pd metal to Ag metal. Furthermore, g-$C_3N_4$ has also been applied in oxidative dehydrogenation of propane (ODHP). According to Chao et al. (2019), g-$C_3N_4$ can be a highly potential metal-free catalyst with high olefin selectivity in selective oxidative dehydrogenation of propane (ODHP) reaction [19]. Other than that, graphitic carbon nitride could be a potential support material for good dispersion of active metal catalysts. For example, addition of

single atom Ru in Ru-g-$C_3N_4$ electrocatalyst showed high photocatalytic hydrogen production rate of 489.7 mmol $g_{Ru}$ $h^{-1}$, which attributed to the addition of single atom of Ru that able to alter electronic structure and improves HER performance, Yu et al., 2022 [20]. While Girma et al. 2021 demonstrated nanocomposite of graphitic carbon nitride and Cu/Fe as catalyst for $CO_2$ reduction to CO with a maximum Faradaic efficiency of 84.4% at a low onset overpotential of $-0.24$ V vs. normal hydrogen electrode (NHE) [21]. In formic acid dehydrogenation, alloying of PdCo supported on g-$C_3N_4$ exhibited synergetic effect and enhanced the catalytic activity that achieved by the effect of alloying, small NPs size, and N-functionalities of the g-$C_3N_4$ support [22]. Homlamai et al. (2020) was also reported the utilization of non-noble metal can contribute to the formation of formate intermediate and needs higher temperature to reduce into $CO_2$ and $H_2$ [6]. Most of the catalytic decomposition of formic acid performed at fast rate and inhibited by formation of the formates species and it was irreversible.

In this study, heterogeneous catalysts especially single metallic catalysts will be synthesized and selected from noble metal and non-noble transition elements such as Pd, Cu, and Zn incorporated with graphitic carbon nitride (g-$C_3N_4$) as support materials. The synthesized catalysts will be characterized and further evaluated over the decomposition of short chain carboxylic acid such as formic acid, acetic acid, and mixture of them to form hydrogen gas. Performance study will be carried out using several parameters to optimize hydrogen production volume.

## 2. Materials and Methods

### 2.1. Materials and Reagents

Palladium (II) chloride ($\geq$99.9 purity) precursor, zinc nitrate hexahydrate (reagent grade, 98%), copper (II) nitrate trihydrate (puriss. p.a., 99–104%), formic acid reagent (ACS reagent, $\geq$96%), melamine (99%) as graphitic carbon nitride source, sodium hydroxide (reagent grade, $\geq$98%,) pellets (anhydrous), sodium borohydride (Reagent Plus®, 99%), and ethanol (ACS reagent, $\geq$99.5%) was used during washing process. All chemicals were purchased from Sigma Aldrich Company (Darmstadt, Germany).

### 2.2. Synthesis of Modified Graphitic Carbon Nitride Based Monometallic Catalyst

To prepare 1 wt% Pd/g-$C_3N_4$ catalyst, a method by Wang et al. [23] was followed with some modifications. Moreover, 0.0433 g of $PdCl_2$ and 1.98 g of g-$C_3N_4$ was added into 30 mL of distilled water. Then, the resulting suspensions was sonicated for 30 min and stirred for 4 h continuously using hot plate. Some amount of the 0.5M sodium hydroxide was added into the suspension to adjust the pH value at 8 to 9. Moreover, 172 mL of 1.0 mg/mL of sodium borohydride solution was poured into the suspension by droplets using a burette for reducing process where the metal precursor was reduced to metallic phase.

Then, the reduced suspension was re-sonicated for 20 min and the suspension was stirred for 1 h continuously by using a hot plate at 10 °C (using ice cubes) to ensure all the palladium chloride completely reduced to metallic phase of Pd. The resulting single metallic catalyst was filtered using centrifuge and washed with ethanol and distilled water in sequence to remove all the organic substance. Subsequently, the suspension was dried in the oven at 10 °C overnight. All the steps have been repeated to produce other catalysts with different amount of $PdCl_2$ precursor according to the metal loadings of 3 wt% and 5 wt% of Pd over graphitic carbon nitride. The same method was applied to prepare the catalysts with 5 wt% metal loading, namely Cu/g-$C_3N_4$ and Zn/g-$C_3N_4$.

### 2.3. Catalysts Characterization

X-ray diffraction (XRD) diffractogram of the synthesized catalysts were recorded on a Bruker AXS D8 Advance (Karlsruhe, Germany) using a Cu K$\alpha$ radiation source (40 kV, 40 mA). Scans were taken over the 2$\theta$ in range of 10° to 80° and wavelength, $\lambda$ = 0.154 nm. Then, 1 g of sample was placed on the sample holder. The data obtain from the analysis was compared with standard peak data reported by Joint Committee on Powder Diffraction

Standard (JCPDS) to match and identify the phase of the sample. The surface morphology and structural of the catalyst was observed by field emission scanning electron microscopy (FESEM) (Zeiss MERLIN, Oberkochen, Germany).

### 2.4. Catalysts Testing

Moreover, 0.03 g of Pd/g-C$_3$N$_4$ catalysts at different metal loadings were used in the reaction. Then, 1.0 mL of formic acid (98% concentration) was titrated into 5 mL 2-neck round bottle flask which connected with rubber tube as depicted in Figure 1. The experiment was observed for any potential reaction in 10 min to 1 h duration. For every 10 min, the gas trapped in the inverted measuring cylinder was measured based on displacement of water. These steps had repeated using different temperature of 50 °C and 70 °C. Subsequently, the steps were the same using different concentration of formic acid (70%, 50%, and 50%–50% formic acid-acetic acid mixture). The gas composition for the optimum reaction condition has been analyzed using gas chromatography (GC-TCD).

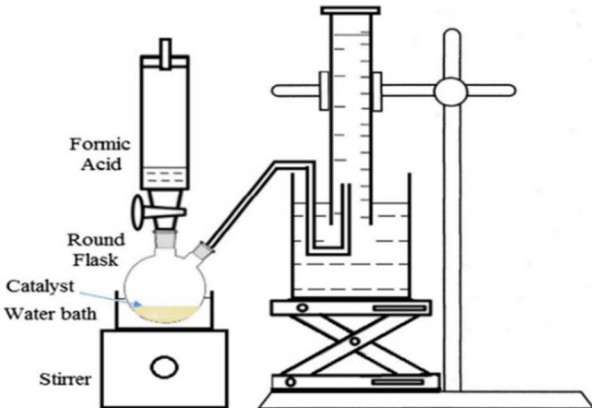

**Figure 1.** Experimental set-up of carboxylic acid decomposition.

The last step is by repeating all the steps above using other catalysts, Cu/g-C$_3$N$_4$ and Zn/g-C$_3$N$_4$, according to the best suitable mass percentage chosen from (1 wt%, 3 wt%, and 5 wt%) Pd/g-C$_3$N$_4$. There are several parameters have been studied in this work such as type of active metal loadings (Pd/g-C$_3$N$_4$, Cu/g-C$_3$N$_4$, and Zn/g-C$_3$N$_4$), percentage of the metal loadings (1 wt%, 3 wt%, and 5 wt%), reactant concentration (98%, 70%, 50% formic acid, and 50%–50% of formic acid-acetic acid solution) and reaction temperature (30, 50, and 70 °C). Dilution of formic acid and acetic acid, TON, and TOF can be calculated using the equation below. The number of available Pd atoms was obtained with EDX analysis.

$$Formic\ acid\ or\ acetic\ acid = \frac{Desired\ dilution\ (\mathrm{mL})}{Initial\ dilution\ (\mathrm{mL})} \times total\ volume\ of\ solution\ (\mathrm{mL})$$

$$Distilled\ water\ volume\ = Total\ volume\ of\ solution - formic\ or\ acetic\ acid\ volume$$

$$\mathrm{TON} = \frac{[Mole\ of\ reactant \times Conversion]}{Moles\ of\ Pd\ NP\ (n)}$$

$$\mathrm{TOF} = \frac{\mathrm{TON}}{Time\ of\ reaction\ (\mathrm{h})}$$

### 3. Results and Discussion

#### 3.1. Catalysts Properties and Features

Five samples of catalysts had been synthesized using Pd, Zn, and Cu metals combined with graphitic carbon nitride which act as support. The Pd/g-C$_3$N$_4$ catalyst was synthesized in three mass percentage ratios (1 wt%, 3 wt%, and 5 wt% of metal in graphitic carbon nitride). For Cu/g-C$_3$N$_4$ and Zn/g-C$_3$N$_4$ catalysts, both were synthesized in a single mass

percentage ratio (5 wt% of metal in graphitic carbon nitride). In addition, a control sample also included which consist of only g-C$_3$N$_4$.

For XRD analysis, all five samples of catalyst were analyzed. For all (1 wt%, 3 wt%, and 5 wt%) Pd/g-C$_3$N$_4$ catalyst, Figure 2 shows that all three Pd/g-C$_3$N$_4$ catalysts have the same highest XRD peak of graphitic carbon nitride (g-C$_3$N$_4$) which can detected between 25° and 30°. For palladium metal element (Pd$^o$), all the three Pd/g-C$_3$N$_4$ catalysts have the same highest XRD peak which can be detected at 40°. There is also a small amount of alloy compound, palladium oxide (PdO) detected in the analysis. From the three samples, 5 wt% Pd/g-C$_3$N$_4$ catalyst has the highest XRD peak of Pd$^o$ compared to 1 wt% Pd/g-C$_3$N$_4$ and 3 wt% Pd/g-C$_3$N$_4$ catalyst.

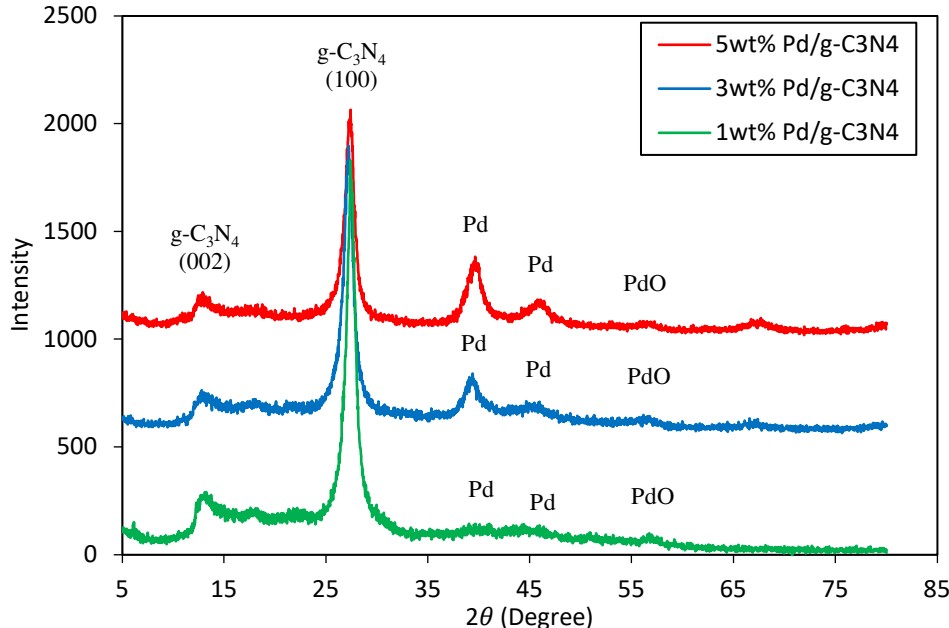

**Figure 2.** XRD diffractogram of Pd/g-C$_3$N$_4$ catalyst at different metal loadings.

For 5 wt% Cu/g-C$_3$N$_4$ catalyst, Figure 3 shows that highest XRD peak of graphitic carbon nitride (g-C$_3$N$_4$) can be detected between 25° and 30°. For copper metal element (Cu°), only few produced and no XRD peak of Cu metal. Instead, there were lot of alloy compound, copper oxide (CuO) can be seen in the analysis. The highest XRD peak for CuO can be detected between 35° and 40°.

For 5 wt% Zn/g-C$_3$N$_4$ catalyst, Figure 4 shows that highest XRD peak of graphitic carbon nitride (g-C$_3$N$_4$) can be detected between 25° and 30°. There are significant peaks represented to Zn$^{2+}$ as ZnO have been detected without any metallic phase of Zn (Zn0). The highest XRD peak for ZnO can be detected between 35° and 40°.

Table 1 shows the percentage of crystallinity and amorphous as well as crystalline size of all synthesized catalyst. The average crystallite size of graphitic carbon nitride supported Pd catalyst ranges from 8 to 17 nm. Higher loadings of Pd gives higher crystallinity of 56.8%. The catalyst with 5 wt% of Zn has bigger average crystal size of 25 nm. It could contribute to the less dispersion of the particles towards less catalytic activity. It was comparable with the previous study that the average crystal size of activated carbon-based Pd catalyst was obtained within the range of 2 nm to 10 nm [23].

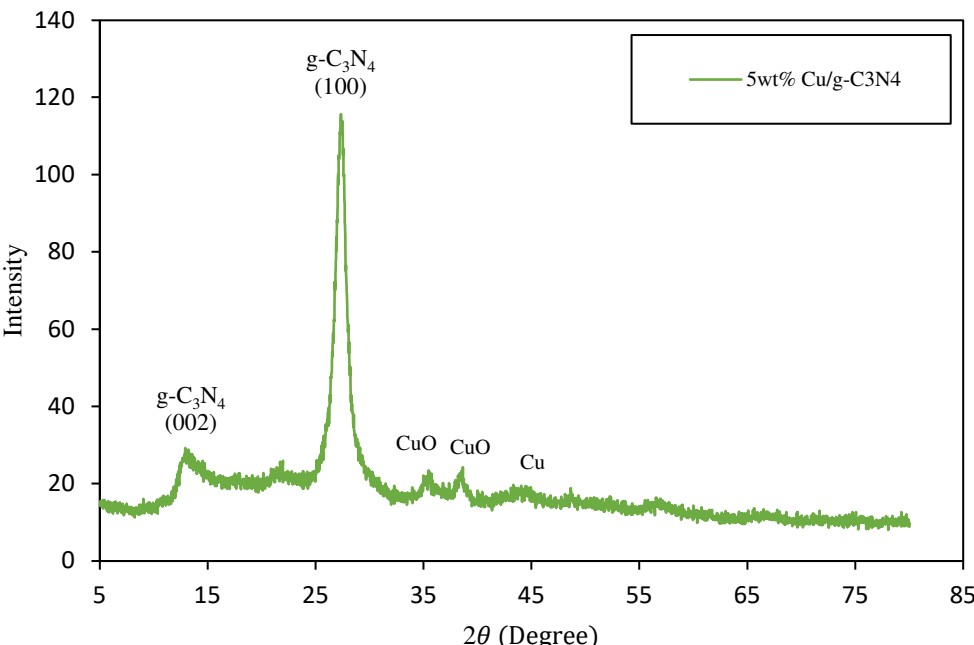

**Figure 3.** XRD analysis of 5 wt% Cu/g-C$_3$N$_4$ catalyst.

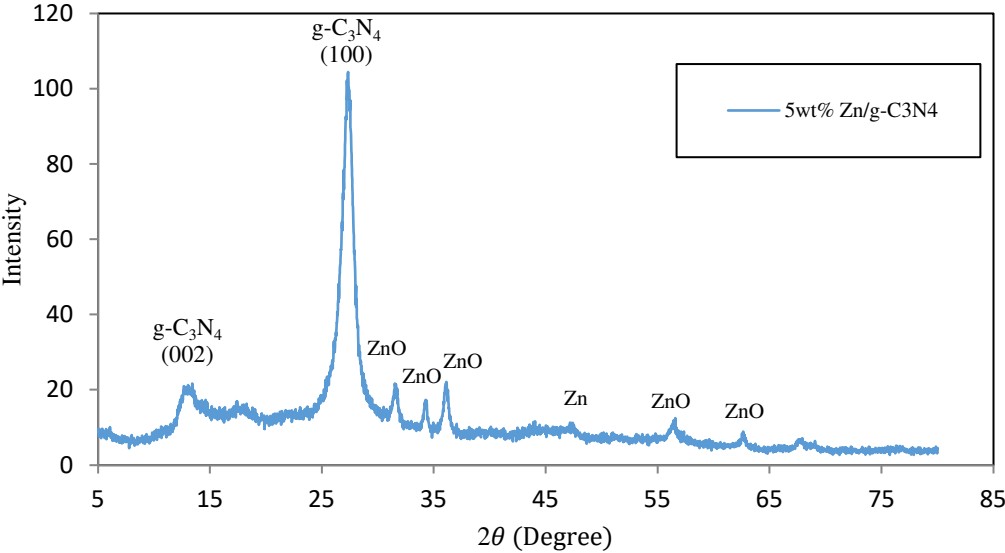

**Figure 4.** XRD analysis of 5 wt% Zn/g-C$_3$N$_4$ catalyst.

**Table 1.** Percentage of crystallinity and amorphous as well as crystallite size of all synthesized catalyst.

| Type of Catalyst | Crystallinity Percentage (%) | Amorphous Percentage (%) | Average Crystal Size of Metal/Metal Oxide (nm) |
|---|---|---|---|
| 1 wt% Pd/g-C$_3$N$_4$ | 47.4 | 52.6 | 8.14 |
| 3 wt% Pd/g-C$_3$N$_4$ | 49.2 | 50.8 | 11.14 |
| 5 wt% Pd/g-C$_3$N$_4$ | 56.8 | 43.2 | 17.63 |
| 5 wt% Cu/g-C$_3$N$_4$ | 48.5 | 51.5 | 20.39 |
| 5 wt% Zn/g-C$_3$N$_4$ | 50.7 | 49.3 | 25.24 |

From physisorption study using BET method, it was found that higher loadings of Pd decrease the surface area of catalyst. Despite the low surface area of carbon nitride, it has several advantages as catalyst support. Its abundant tri-s-triazine groups can serve as anchoring points for metal precursors to produce small and well-dispersed NPs [22]. Apart from that, the rich electrons of nitrogen atoms in g-$C_3N_4$ support could transfer to Pd NPs and increase the electron cloud density of Pd to promote dissociation of hydrogen atom [24]. Surface properties analysis using BET method are summarized in Table 2.

**Table 2.** Surface properties analysis using BET technique.

| Catalysts | BET Surface Area (m²/g) | Micropore Area (m²/g) | Pore Volume (cm³/g) | Pore Size (nm) |
|---|---|---|---|---|
| GCN | 8.56 | 1.189 | 0.0196 | 9.14 |
| 1% Pd/GCN | 7.55 | 1.76 | 0.0172 | 9.12 |
| 3% Pd/GCN | 7.32 | 1.62 | 0.0173 | 9.45 |
| 5% Pd/GCN | 7.18 | 1.35 | 0.0193 | 9.45 |
| 5% Cu/GCN | 10.27 | 1.77 | 0.0226 | 8.82 |
| 5% Zn/GCN | 10.25 | 1.54 | 0.0215 | 8.41 |

For FESEM analysis, it only can be done for all three types of Pd/g-$C_3N_4$. Based on the EDX analysis, 1 wt% Pd/g-$C_3N_4$ has composition of 64.9 wt% nitrogen, 30.9 wt% carbon, 3.3 wt% oxygen, and 0.9 wt% palladium. Figure 5 shows the surface morphology of the catalyst at magnification of 10.00 KX. Based on the morphology analysis of all three catalyst of Pd/g-$C_3N_4$, the Pd particles are successfully dispersed (white spot) over the graphite-carbon nitride support with small crystal formation by using modification method from Wang et al. [23]. In FESEM images of each catalyst, graphitic carbon nitride in each catalyst samples also presence in layered structure, indicating the graphitic carbon nitride consisted of graphitic planes stacking along the c-axis, as also reported by previous literature [25].

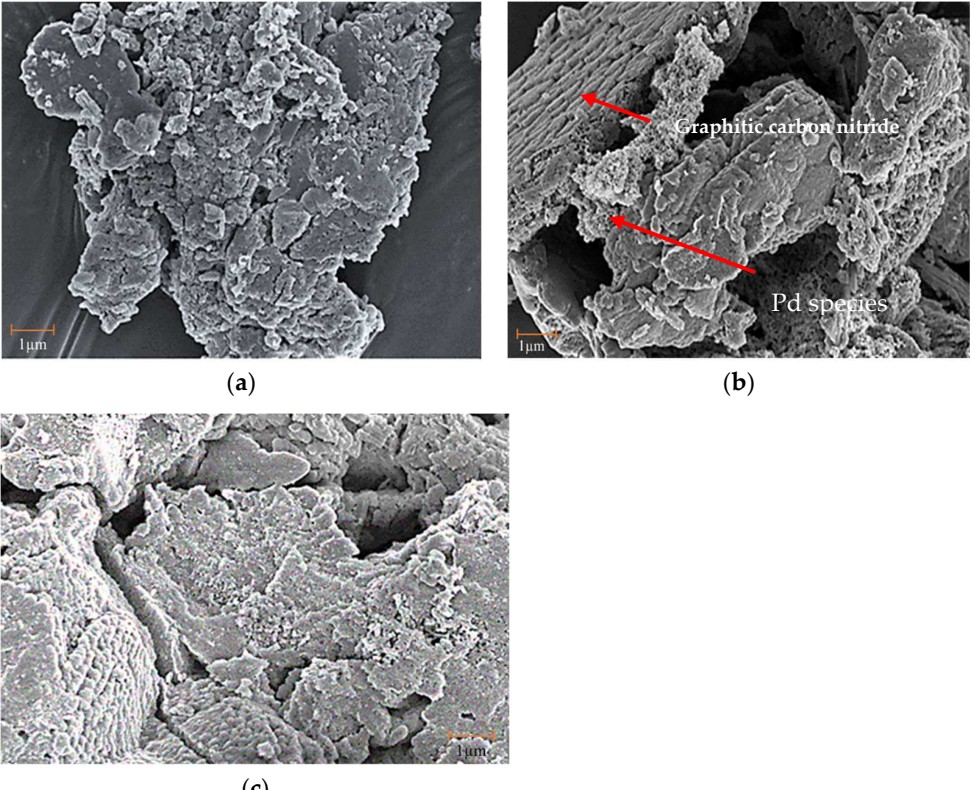

**Figure 5.** Surface morphology of (**a**) 1 wt% Pd/g-$C_3N_4$, (**b**) 3 wt% Pd/g-$C_3N_4$, and (**c**) 5 wt% Pd/g-$C_3N_4$ at magnification of 10.00 KX.

Based on the EDX analysis, 3 wt% Pd/g-C$_3$N$_4$ has composition of 63.2 wt% nitrogen, 29.1 wt% carbon, 5.4 wt% oxygen, and 2.3 wt% palladium. Figure 6 shows the EDX analysis of the samples with different Pd loadings and have been summarized in Table 3.

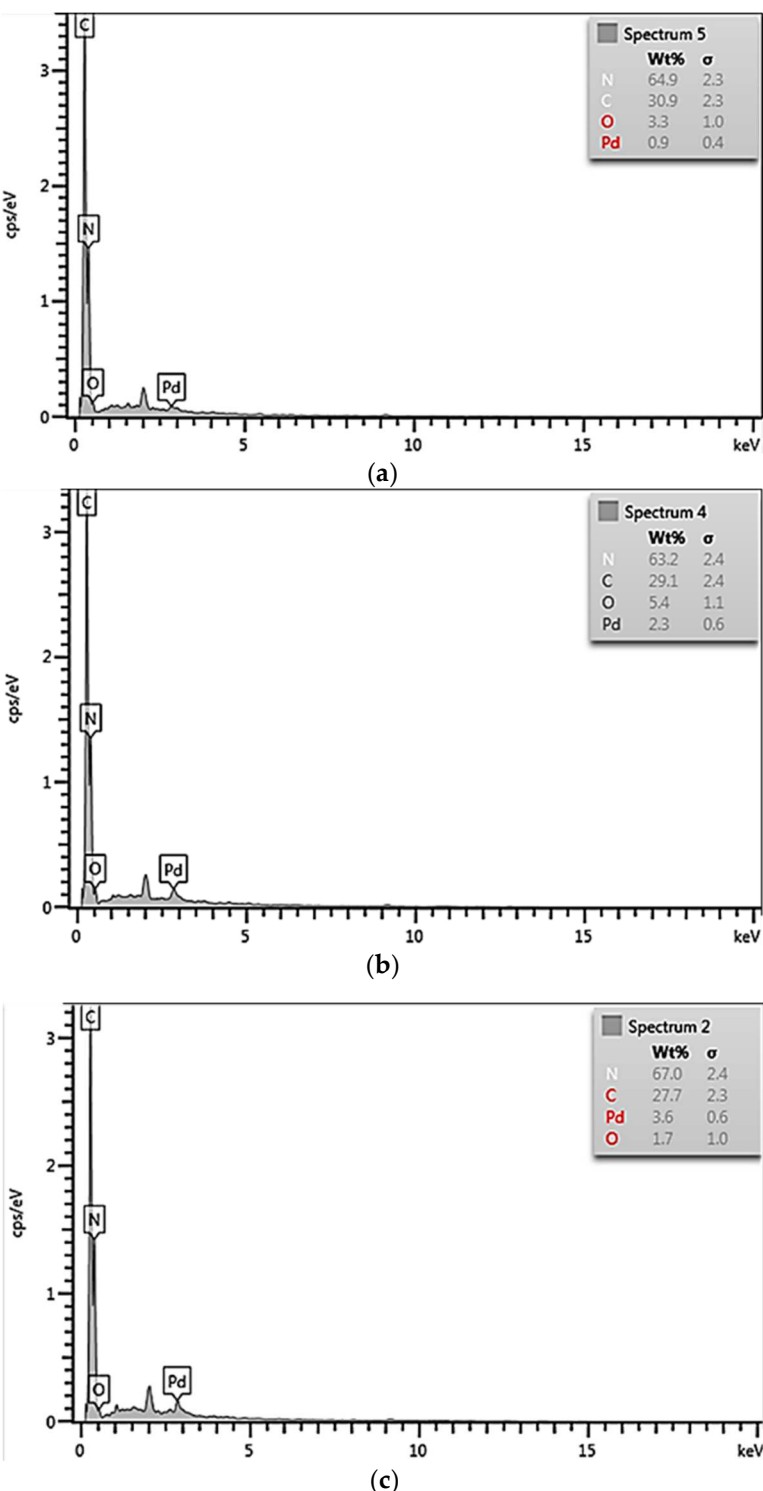

**Figure 6.** EDX analysis of the catalyst samples, (**a**) 1 wt% Pd/ g-C$_3$N$_4$, (**b**) 3 wt% Pd/g-C$_3$N$_4$, and (**c**) 5 wt% Pd/g-C$_3$N$_4$.

**Table 3.** Summarized of EDX analysis results of the samples at different Pd loadings.

| Elements | Catalysts with Different Pd Loadings (wt%) | | |
| --- | --- | --- | --- |
| | **1 wt% of Pd** | **3 wt% of Pd** | **5 wt% of Pd** |
| Nitrogen | 64.9 | 63.2 | 67.0 |
| Carbon | 30.9 | 29.1 | 27.7 |
| Oxygen | 3.3 | 5.4 | 1.7 |
| Palladium | 0.9 | 2.3 | 3.6 |

Based on the EDX analysis, 5 wt% Pd/g-$C_3N_4$ has composition of 67.0 wt% nitrogen, 27.7 wt% carbon, 1.7 wt% oxygen, and 3.6 wt% palladium. Small amounts of oxygen content in the catalysts may represent the oxygenated phase of active metal as PdO. From the morphological analysis, Pd particles are widely dispersed over graphitic carbon nitride and all three catalysts have some small crystal formation on the surface of support material.

*3.2. Catalytic Performance over Formic Acid, Acetic Acid and their Mixture Decomposition Reaction*

Two parameters that are involved for the optimization in decomposition reaction are formic acid concentration, decomposition temperature and Pd loadings, mixture of formic acid-acetic acid, and effect of active metals, namely Pd, Cu, and Zn. For concentration of formic acid and mixture of formic acid-acetic acid, three types of catalysts were reacted with the formic acid at different concentration of 98%, 70%, and 50%, as well as a mixture of 50%–50% formic acid and acetic acid at constant temperature of 30 °C. Figure 7 shows 1 wt% Pd/g-$C_3N_4$ catalyst showed a reaction with all formic acid concentration. At higher concentration of 98% formic acid, the catalyst showed a better performance than lower concentrations. The active sites of the Pd surface were saturated with increasing formic acid concentrations, which might result in a decrease in hydrogen generation. Similar results were reported for formic acid decomposition to produce hydrogen by Mihet et al., 2020 [26] and Sanchez et al., 2018 [27].

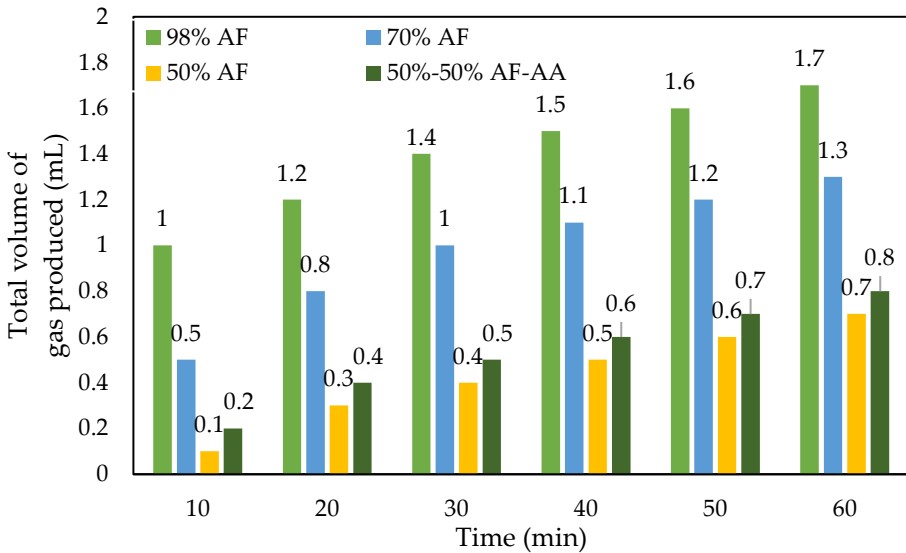

**Figure 7.** Decomposition of formic acid (98%, 1 mL) and the mixture of formic-acetic acid (50%:50%, 1 mL) at different concentration using 1 wt% Pd/g-$C_3N_4$ at 30 °C.

Based on Figure 8, 1 wt% Pd/g-$C_3N_4$ catalyst was tested in 98% formic acid at 30 °C, 50 °C, and 70 °C. The catalyst performed well at 70 °C by producing gas ($H_2$ and $CO_2$), approximately of 4.3 mL which is much higher than the reaction at 30 °C and 50 °C. High temperature was increased the movement of the molecules and weaken the intramolecular bonds of formic acid. The presence of catalyst accelerating the C-H and o-H bonds breakout to form $H_2$ and $CO_2$.

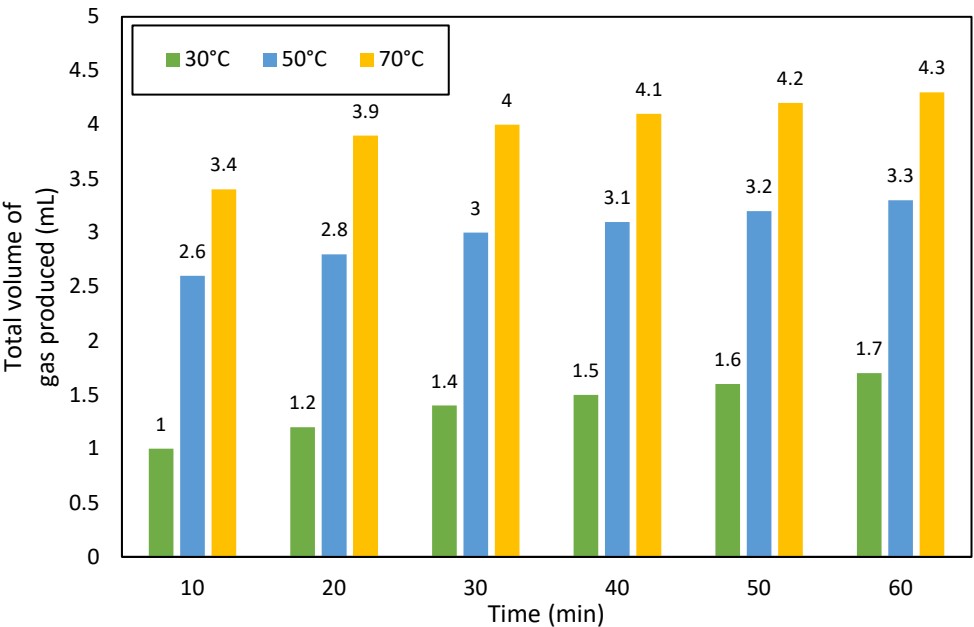

**Figure 8.** Decomposition of 98% formic acid (1 mL) at different temperature over 1 wt% Pd/g-C$_3$N$_4$ catalyst.

Figure 9 shows 3 wt% Pd/g-C$_3$N$_4$ catalyst react with all types of concentration and at concentration 98% formic acid, and the catalyst shows better performance than other concentrations. Thus, 98% concentration of formic acid has been selected for further evaluation by other reaction parameters such as temperature and different Pd metal loadings as shown in Figures 10 and 11.

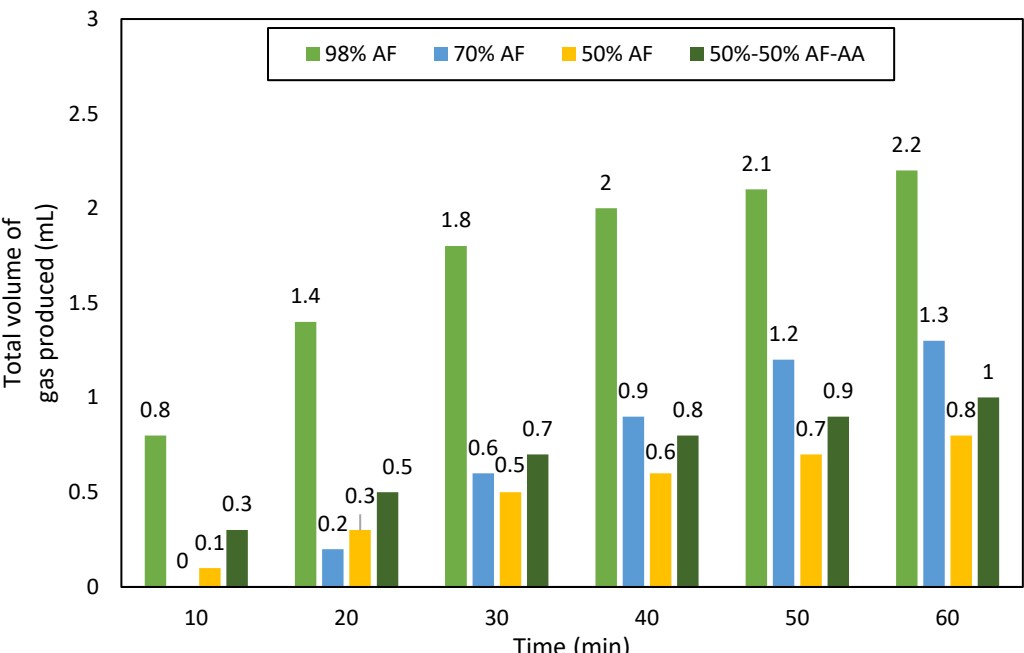

**Figure 9.** Decomposition of formic acid (98%, 1 mL) and mixture of formic acid-acetic (50%:50%, 1 mL) at different concentration over 3 wt% Pd/g-C$_3$N$_4$ catalyst at 30 °C.

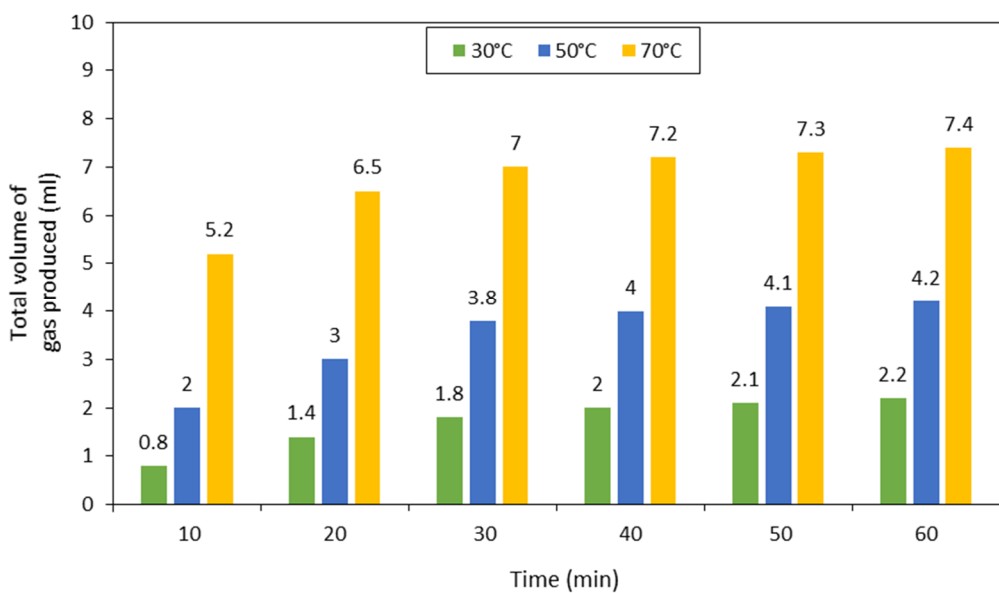

**Figure 10.** Decomposition of 98% formic acid (1 mL) at different temperature over 3 wt% Pd/g-C$_3$N$_4$ catalyst.

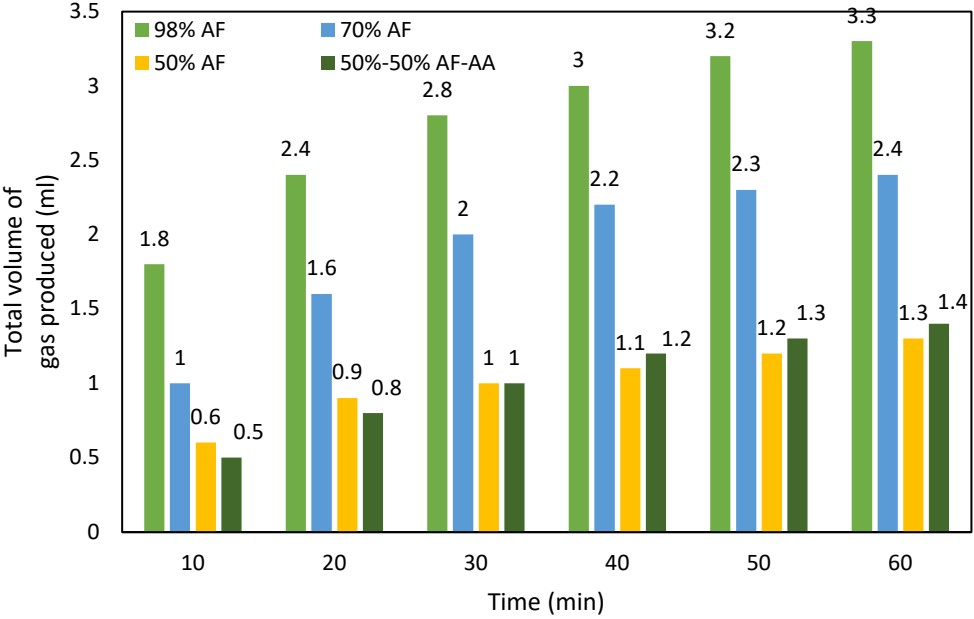

**Figure 11.** Decomposition of formic acid (98%, 1 mL) and mixture of formic acid-acetic (50%:50%, 1 mL) at different concentration over 5 wt% Pd/g-C$_3$N$_4$ catalyst (0.03 g) at 30 °C.

Based on Figure 10, 3 wt% Pd/g-C$_3$N$_4$ catalyst was tested in 98% formic acid at 30, 50, and 70 °C. The catalyst performed well at 70 °C by producing gas (H$_2$ and CO$_2$), approximately 7.4 mL more than temperature at 30 and 50 °C. Figure 10 shows 5 wt% Pd/g-C$_3$N$_4$ catalyst react with all types of concentration and at concentration 98% formic acid, the catalyst shows better performance than other concentrations. Thus, 98% concentration of formic acid has been fixed for the second parameter which is temperature.

Based on Figure 12, 5 wt% Pd/g-C$_3$N$_4$ catalyst was tested in 98% formic acid at 30, 50, and 70 °C. The catalyst performed well at and 70 °C by producing gas (H$_2$ and CO$_2$), approximately 3.3 mL more than temperature at 30 and 50 °C.

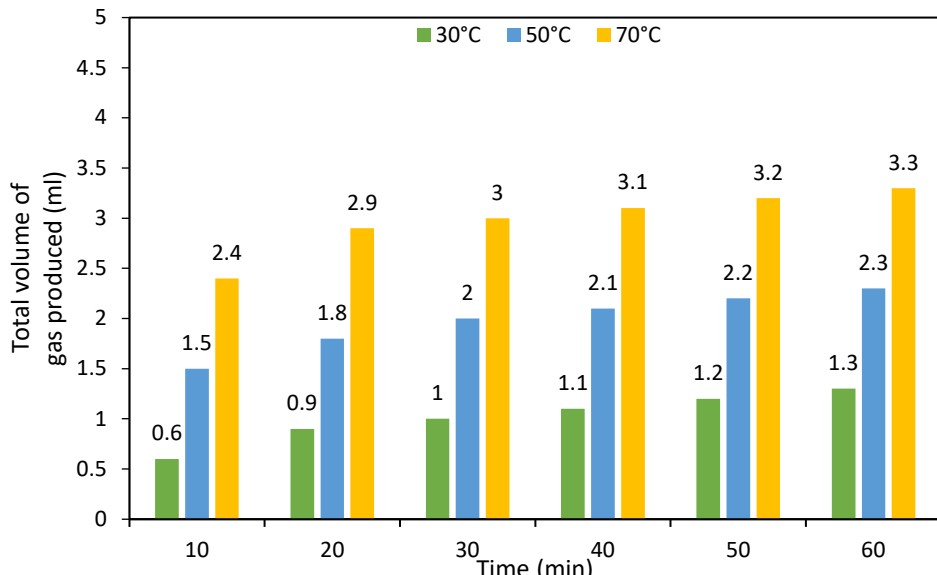

**Figure 12.** Decomposition of 98% formic acid (1 mL) at different temperature using 5 wt% Pd/g-C₃N₄ catalyst.

Based on Figures 13–15, it shows that 5 wt% Pd/g-C₃N₄ performed better compared to other catalyst. Moreover, 5 wt% Pd/g-C₃N₄ produced ($H_2$ and $CO_2$) much higher than other catalyst in all concentrations and temperatures. This may be due to the higher content of Pd metal in the 5 wt% Pd/g-C₃N₄ compared to other catalyst. It has been reported that Pd metal is the most active element or monometallic element in ambient condition and Pd metal also performed well in catalytic activity [28]. Thus, having high percentage of mass of Pd metal, the performance of decomposition reaction will increase. However, for higher temperature up to 70 °C, it showed a different trend while the total gas was decreased for 5 wt% Pd/C₃N₄ catalyst. It may be due to the early dehydration reaction of formic acid to produce PdCO or Pd formate caused the inhabitation of the catalyst [29].

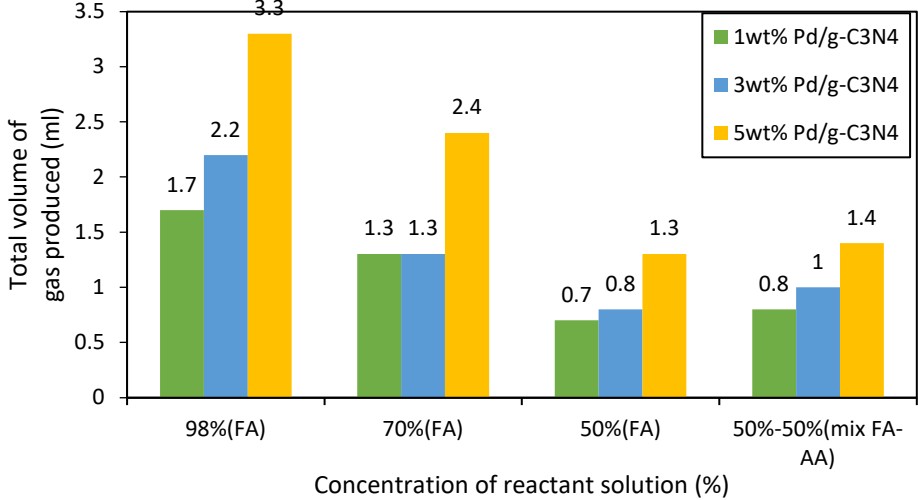

**Figure 13.** Decomposition of formic acid (98%, 1 mL) and mixture of formic acid-acetic (50%:50%, 1 mL) at different concentration using (1 wt%, 3 wt%, and 5 wt%) Pd/g-C₃N₄ at 30 °C after 60 min.

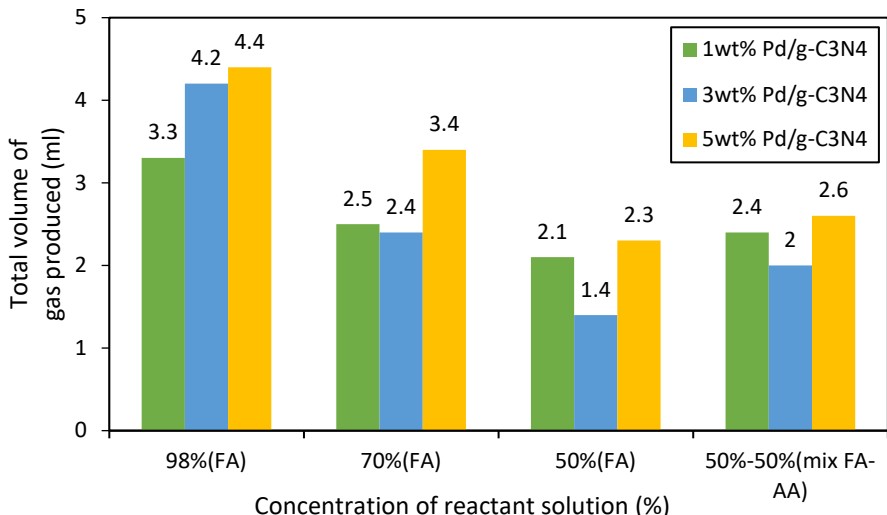

**Figure 14.** Decomposition of formic acid (98%, 1 mL) and mixture of formic acid-acetic (50%:50%, 1 mL) at different concentration using (1 wt%, 3 wt%, and 5 wt%) Pd/g-C$_3$N$_4$ at 50 °C after 60 min.

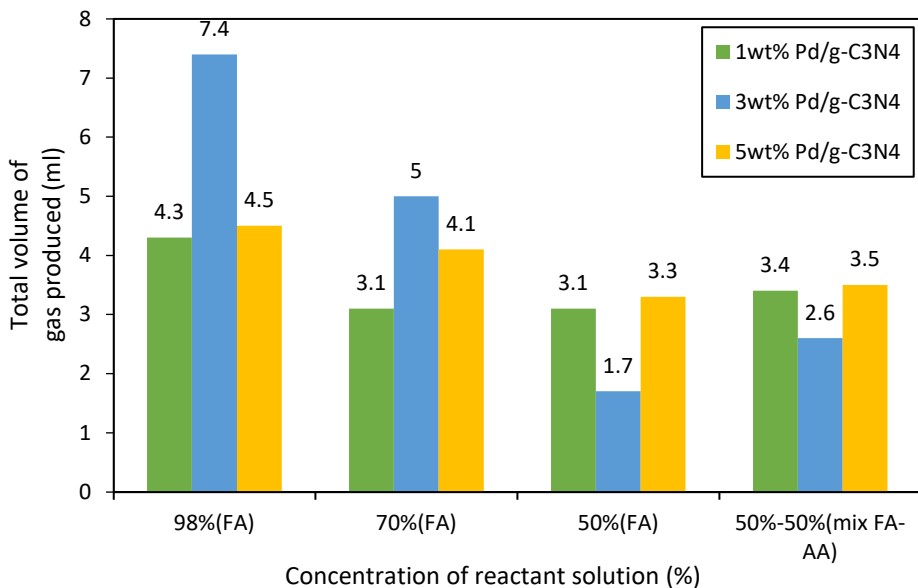

**Figure 15.** Decomposition of formic acid (98%, 1 mL) and mixture of formic acid-acetic (50%:50%, 1 mL) at different concentration using (1 wt%, 3 wt%, and 5 wt%) Pd/g-C$_3$N$_4$ at 70 °C for 60 min.

Based on Figure 16a, it shows the conversion rate for 5 wt% Pd/g-C$_3$N$_4$ catalyst in decomposition reaction using 98% formic acid at different temperature (30 °C, 50 °C, and 70 °C) by assuming that 1 mol of formic acid produce 1 mol of gas (H$_2$ and CO$_2$) at STP. The highest conversion rate achieved by 5 wt% Pd/g-C$_3$N$_4$ catalyst is 0.741% at 70 °C. It showed significant value of formic acid conversion than other reported works from several researchers. Mihet et al. (2020) reported that templated carbon supported Pd catalyst did not showed any conversion of formic acid at room temperature [26]. The 5 wt% Pd/g-C$_3$N$_4$ catalyst was considered the best performance catalyst based on total gas production at room temperature. Since the higher temperature up to 70 °C did not show very significant in gas production, performance at room temperature might be more economical and practical. The optimum performance of 5 wt% Pd/g-C$_3$N$_4$ was shown at room temperature, compared with higher temperature of 50 and 70 °C. It indicated that the Pd atoms were stable at low temperature and less stable at higher temperature to produce possible interaction with formate and created a passive layer towards decreasing in the catalytic performance [24]. The measurement of kinetics was conducted over a range of temperature of 30 to 70 °C

for 5 wt% Pd/g-C$_3$N$_4$. The Arrhenius plot in Figure 16b showed the activation energy was calculated to be 19.81 kJ/mol with the R$^2$ of 0.986, indicating the kinetic equation was reasonable for the reaction.

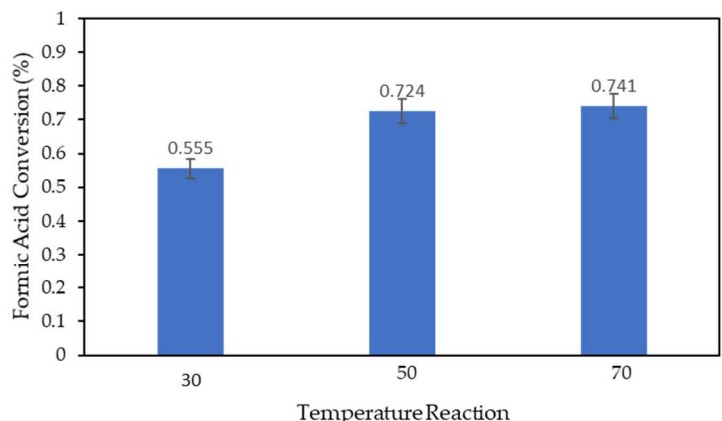

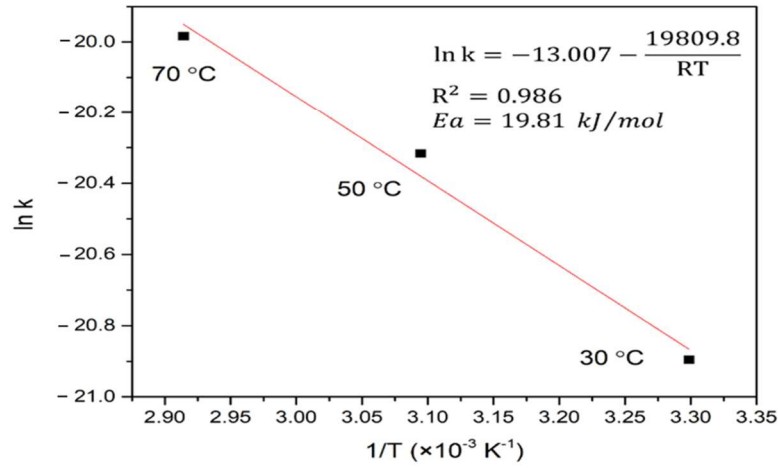

**Figure 16.** (**a**) Conversion rate for 5 wt% Pd/g-C$_3$N$_4$ catalyst in decomposition reaction using 98% formic acid (1 mL) at different temperature (30 °C, 50 °C, and 70 °C) for 60 min. (**b**) Arrhenius plot of the reaction.

In Table 4, a summarized results of catalysts' performance from this study and other researchers is shown. In this study, the highest TOF$^{-1}$ achieved is 1449.3 h$^{-1}$, by 5 wt% Pd/g-C$_3$N$_4$ catalyst, while Cu and Zn based catalysts achieved significantly low TOF of 57.2 and 23.2 h$^{-1}$, respectively. It was comparable with other findings [30,31] using Pd based heterogeneous catalysts. Highest Pd content of 5 wt% can provide more active site with most of the g-C$_3$N$_4$ support surface which has been covered with Pd species as shown in FESEM images (Figure 4). In heterogeneous catalysis, the highest TOF achieved is 7256 h$^{-1}$, by palladium nanoparticles immobilized on carbon nanospheres developed by Zhu et al. [32]. High dispersion of fine Pd active metal contributed to the highest activity.

**Table 4.** Conversion, turnover frequency (TOF) and turnover number (TON) for 5 wt% Pd/g-C$_3$N$_4$, 5 wt% Cu/g-C$_3$N$_4$, and 5 wt% Zn/g-C$_3$N$_4$ catalyst (0.03 g) in decomposition reaction using 98% formic acid (1 mL) at 30 °C for 60 min.

| Catalysts | FA Conversion (%) | TON | TOF (h$^{-1}$) | Total Gas Produced (mL) (1st Cycle) | Total Gas Produced (mL) (2nd Cycle) | Total Gas Produced (mL) (3rd Cycle) |
|---|---|---|---|---|---|---|
| 5 wt% Pd/g-C$_3$N$_4$ | 0.555 | 1449.3 | 1449.3 | 3.3 | 3.0 | 2.7 |
| 5 wt% Cu/g-C$_3$N$_4$ | 0.051 | 57.2 | 57.2 | 0.3 | n.d | n.d |
| 5 wt% Zn/g-C$_3$N$_4$ | 0.020 | 23.2 | 23.2 | 0.12 | n.d | n.d |
| Pd/rGO [30] | n.d | n.d | 911.0 (10 mL of FA used) | 80 | n.d | n.d |
| Pd/C [31] | 9.5 | n.d | 1500 (at 50 °C, 5 bar) | n.d | n.d | n.d |
| IrCp * Cl2bpym [33] | n.d | n.d | 2490 (at 40 °C, 300 mL of FA) | 350 | n.d | n.d |

n.d: Not determined; * denotes the presence of chiral carbon in the compound; Pd, Cu, and Zn atomic weight were estimated using EDX based on 0.03 g of total catalysts used.

The decomposition reaction is continued for the control sample (g-C$_3$N$_4$) and other catalysts (5 wt% Cu/g-C$_3$N$_4$ and 5 wt% Zn/g-C$_3$N$_4$) by using 98% formic acid at temperature of 30 °C. Figure 17 shows the decomposition reaction of 98% formic acid with 5 wt% Pd/g-C$_3$N$_4$, 5 wt% Cu/g-C$_3$N$_4$, and 5wt% Zn/g-C$_3$N$_4$ at 30 °C. From Figure 16 below, 5 wt% Pd/g-C$_3$N$_4$ shows better performance compared to both of other catalyst. Moreover, 5 wt% Pd/g-C$_3$N$_4$ managed to produce gas about 3.3 mL compared to 5 wt% Cu/g-C$_3$N$_4$ and 5 wt% Zn/g-C$_3$N$_4$ as they produce very little gas. This may be due to the large formation of metal oxide formed based on the XRD analysis. The presence of metal oxide phase on the catalysts such as CuO and ZnO were recognized as less active species for the decomposition of carboxylic acid and could produce less gaseous products. The collected gas for 5 wt% metal loading catalysts have been analyzed using GC-TCD and the results are summarized in Table 5. Moreover, 5 wt% Pd/g-C$_3$N$_4$ showed highest hydrogen content up to 95.3% compared with Cu and Zn based catalysts. Lower hydrogen content due to higher CO$_2$ production through formic acid decomposition. Meanwhile, no other gas such as carbon monoxide and methane has been detected.

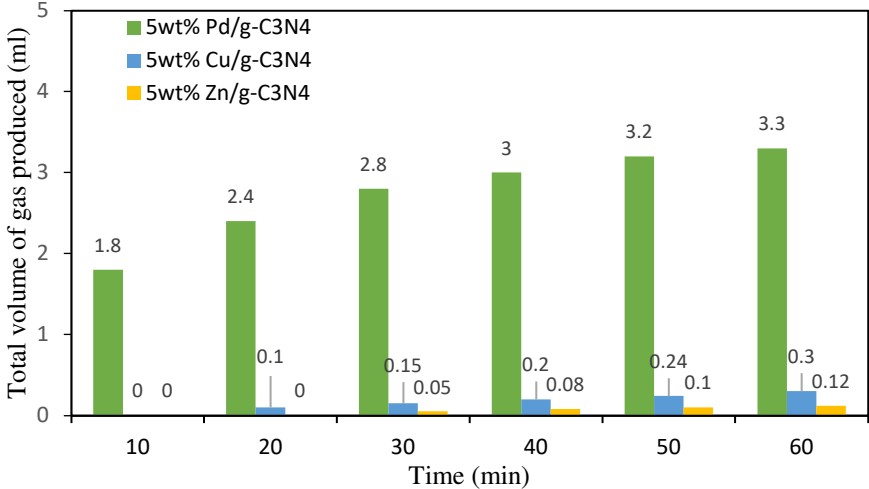

**Figure 17.** Decomposition of 98% formic acid using 5 wt% Pd/g-C$_3$N$_4$, 5 wt% Cu/g-C$_3$N$_4$, and 5 wt% Zn/g-C$_3$N$_4$ at 30 °C.

**Table 5.** Gas composition analysis using GC-TCD for formic acid decomposition over different catalysts at 30 °C after 60 min.

| Catalysts | Hydrogen (%) | Carbon Dioxide (%) | Other Gas ($N_2$) (%) |
|---|---|---|---|
| 5 wt% Pd/g-$C_3N_4$ | 95.3 | 4.7 | n.d |
| 5 wt% Cu/g-$C_3N_4$ | 92.0 | 6.7 | 1.3 |
| 5 wt% Zn/g-$C_3N_4$ | 87.5 | 10.5 | 2.0 |

n.d: Not detected.

## 4. Conclusions

Graphitic carbon nitride (g-$C_3N_4$) based materials have the potential to decompose short chain carboxylic acid into hydrogen gas. The effect of active metals such as Pd, Zn, and Cu gives an interesting finding that can be explored in the future. Reaction temperature showed a significant effect for formic acid-acetic acid decomposition. The optimum concentration and temperature for the catalyst to perform well are 98% concentration of formic acid and 70 °C, respectively. The g-$C_3N_4$ based catalysts with 5 wt% loading of the Pd metal is the most active catalyst compared with other catalysts with the highest gas production of 3.3, 4.4 and 4.5 mL at 30, 50, and 70 °C, respectively. Decomposition at room temperature gives additional advantages since no external heat sources are needed and it is much more practical to apply at any sites.

**Author Contributions:** Conceptualization, W.N.R.W.I.; methodology, M.N.K.; validation, W.N.R.W.I. and A.A.-A.; formal analysis, K.N.A. and Z.A.C.R.; investigation, M.N.K.; resources, W.N.R.W.I.; data curation, W.K.A.-A.; writing—original draft preparation, M.N.K. and W.N.R.W.I.; writing—review and editing, A.A.-A., Z.A.C.R., and W.N.R.W.I.; visualization, K.N.A.; supervision, W.N.R.W.I. and A.A.-A.; funding acquisition, W.N.R.W.I. All authors have read and agreed to the published version of the manuscript.

**Funding:** This research work was funded by the Ministry of Higher Education of Malaysia, and Universiti Kebangsaan Malaysia (UKM) under research code: FRGS/1/2020/TK0/UKM/02/31 and GUP-2020-012, respectively.

**Institutional Review Board Statement:** Not applicable.

**Informed Consent Statement:** Not applicable.

**Data Availability Statement:** Not applicable.

**Acknowledgments:** The authors wish to acknowledge Ministry of Higher Education for the financial support under research code: FRGS/1/2020/TK0/UKM/02/31, Universiti Kebangsaan Malaysia (UKM) for the financial support under research code: GUP-2020-012, and Centre for Research & Instrumentation Management (CRIM) for the support.

**Conflicts of Interest:** The authors declare no conflict of interest.

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
