# Peer review of "Decomposition of Formic Acid and Acetic Acid into Hydrogen Using Graphitic Carbon Nitride Supported Single Metal Catalyst"

_sustainability, doi:10.3390/su142013156_

Round 1
Reviewer 1 Report
The authors describe catalysts as well as an experimental setup to measure decomposition of formic acid into H2 and CO2. The reaction was performed at different temperatures (30-70°C) on acid solutions with different concentrations. Conversion was monitored by volumetric measurement of gas formation. The paper has several systematic weaknesses:
Catalysts were prepared from PdCl2 as precursor and NaOH (I guess that is what is meant by "Sodium hydrochloride" in line 93?) Was used for pH adjustment. How did the authors ensure that the resulting catalysts are Cl and Na free? Both elements can have an effect on catalytic performance.
What is the rational to use carbon nitride as support (especially since the surface area is so low - the authors report values even < 10m²/g)? What is the advantage over e.g. a high surface area carbon? Is a control sample available, demonstrating this benefit? Why is it necessary to use a supported catalyst at all? Is there a benefit over metal particles formed in situ.
How do the results of the current work compare to other state of the art systems (e.g. Fellay, Céline; Dyson, Paul J.; Laurenczy, Gábor (2008). "A Viable Hydrogen-Storage System Based on Selective Formic Acid Decomposition with a Ruthenium Catalyst". Angewandte Chemie International Edition. 47 (21): 3966–8. doi:10.1002/anie.200800320. or Onishi N, Kanega R, Kawanami H, Himeda Y. Recent Progress in Homogeneous Catalytic Dehydrogenation of Formic Acid. Molecules. 2022 Jan 11;27(2):455. doi: 10.3390/molecules27020455. PMID: 35056770; PMCID: PMC8781907 for a review of homogeneous systems).
Why are catalysts added with the same total amount and therefore varying Pd content in each experiment? The outcome is trivial: more Pd in the reactor should result in higher gas formation. Wouldn't a comparison based on the same amount of Pd but with different concentration on the support give more interesting results as it would probe for Pd dispersion effects?
Did the authors analyse the composition of the gas formed in the reaction? Is the H2/CO2 ratio constant at all conditions or is there evidence for some CO2 dissolution in the reaction mixture or in the water used in the gas trapping apparatus? How can the authors be sure, H2/CO2 is formed at all? Is it possible that CO/H2O is formed instead? If the goal of the hydrogen formation is to use the resulting gas in a fuel cell, even traces of CO are problematic, so the selectivity of the catalyst is critical. Therefore, analysis of the reaction gas (e.g. by FTIR) is mandatory and state of the art.
The finding that reaction rate depends on temperature is trivial and subject of any textbook on chemical kinetics. I would recommend to re-evaluate the available data with respect to reaction rates. Since experiments were perfromed at different temperatures, it would even be possible to estimate the activation energy for each catalyst using an Arrhenius plot.
All results are reported as absolute amount of gas formed. For some plots, there is even no duration reported, in which this amount of gas is formed. Since also the exact experimental conditions (temperature of the gas during measurement) are not reported, it is impossible to relate the reported data to any result from the literature. Usually, the performance of catalysts is reported as a turn over frequency and turn over number.
Figure 14 shows that the 3%Pd-catalyst has a pronounced advantage over the 5%Pd sample. This unexpected finding is not discussed at all. Is it significant?
What is the relevance of the characterisation data reported? Can any of these data contribute to explain the unexpected ranking shown in Figure14? What is the significance of increasing deviation between target Pd-content and Pd content observed by EDX?
Author Response
Dear reviewer,
Thank you for the comments and suggestions. All the comments have been taken into account and revised accordingly in the revised version of the manuscript highlighted.
Best regards

Reviewer 2 Report
Manuscript ID: sustainability-1914643
In the submission titled “Decomposition of Formic Acid and Acetic Acid into Hydrogen using Graphitic Carbon Nitride Supported Single Metal Catalyst”, Isahak et al. reported a study for the development of metal-incorporated graphitic carbon nitride (g-C3N4) and applied the as-synthesized catalyst to the decomposition reactions. The authors performed the systematic characterizations of as-synthesized g-C3N4 by using the various experimental techniques (PXRD, FESEM, BET etc) and demonstrated the optimal decomposition of formic acid with the Pd-incorporated g-C3N4. This field is certainly emerging and thus this study looks interesting. Overall, I recommend the publication of this paper, but the following questions and comments should be addressed by the authors.
(i) In the introduction part, the authors should consider to provide the prospective audience a better background information for the g-C3N4–based catalytic reactions. The following literatures could serve this purpose on some aspects; Appl. Catal. B, 2015, 176–177, 44–52 and Catal. Sci. Technol., 2021, 11, 6401–6410.
(ii) Regarding the “Materials and Methods” section, the author should clearly describe the details about the chemical reagents such as the manufacturer, impurity, and so on.
(iii) Based on the FESEM image, the authors should clearly visualize the scale bar. It is too obscure in the current version.
(iv) According to the results from the catalytic reactions with Pd-incorporated g-C3N4, the increase of reaction temperature up to 70 degree showed the highest yield of gas production. I’m wondering how the author set the maximum temperature for those reactions. What will happen in the temperature above 70 degree?
(v) At least under the optimal catalytic condition, the author should report the chemical stability of Pd-incorporated g-C3N4 based on the reusability test.
- Minor comments –
I found some typos. The author should carefully check the typos in the entire manuscript.
(i) In abstract, “… three catalyst loading of 1 wt%, 3 wt% dan 5 wt% of Pd over g-C3N4 …”
The word of “dan” should be changed to “and”.
(ii) In the section of 3.1, “… 5 wt% Pd/g-C3N4 catalyst have the highest XRD peak of Pd° compared to …”
What means the word of “Pd°” ?
(iii) In the section of 3.2, “The catalyst performed well at 70oC by producing gas (H2 and CO2), …”
The unit of temperature should be correctly changed.
Author Response
Response to the Reviewer Comments
Dear reviewer,
Thank you for the comments and suggestions. All the comments have been taken into account and revised accordingly in the revised version of the manuscript highlighted.
Thank you
Best regards

Reviewer 3 Report
In this manuscript, the authors prepared carbon nitride supported metals as catalysts for decomposition of short chain carboxylic acid into hydrogen gas. Metal species, loading amount and catalytic conditions were carefully optimized. This manuscript was clearly written and can be published after addressing the following concerns.
1) The introduction part is not systematically involved, the modifications of carbon nitride to enrich the active sites (surface area, vacancy, functional groups), facilitate charge transfer efficiency (heterojunctions) should be introduced in brief. (Angew. Chem. Int. Ed. 2013, 52, 11822–11825; https://doi.org/10.1016/j.cej.2021.132388; https://doi.org/10.1016/j.ijhydene.2018.11.057; https://doi.org/10.3390/catal12060672).
2) Detailed TEM images before and after the reaction should be given.
3) More characterizations such as XPS should be given and why the optimized catalysts performed best are highly suggested to discuss in details.
4) The comparison of performance on the optimized catalysts with reported catalysts in literatures is suggested to supply.
5) Formats or errors should be carefully checked including lots of references (such as formats of literature titles).
Author Response
Dear reviewer,
Thank you for the comments and suggestions. All the comments have been taken into account and revised accordingly in the revised version of the manuscript highlighted.
Thank you
Best regards

Reviewer 4 Report
The authors synthesized the monometallic catalysts namely Pd, Cu and Zn on graphitic-carbon nitride (g-C3N4) and studied the potential of the catalysts in formic acid (FA) and mixed formic acid (FA)-acetic acid (AA) decomposition reaction. Several parameters have been studied in this work such as the type of active metals, temperature, and metal loadings. The optimized conditions for different metallic catalysts were found. It was found that the catalytic activity of g-C3N4 loaded with Cu and Zn is lower than their counterpart loaded with Pd, due to the formation of unactive CuO and ZnO. This work is interesting and is suitable for publication in Sustainability after addressing the following points:
1. The abstract is too long, which should be more concise to show the key scientific findings. In addition, some grammar issues and typos are found in the abstract. For example, “will increased”, “was significantly reduce”, etc.
2. The morphologies in the SEM images (Figure 4) should be discuss to show readers more comprehensive information.
3. The EDX figures shown in Figure 5 are very unclear. The authors should provide new figures with higher resolution.
4. Some relevant works can be cited and discussed in the manuscript to attract more readerships (e.g. J. Mater. Sci. Technol. 2022, 104, 155 -162; J. Colloid Interface Sci. 2022, 610, 495-503; J. Mater. Chem. C 2021, 9, 14876 -14884).
Author Response

(The authors gave the same response as above.)

Round 2
Reviewer 1 Report
The modifications improved the paper to some extent, however in my opinion this work is still incomplete. The authors included TON and TOF for some samples, however the most interesting relation TON/TOF as function of Pd content, which would allow to draw conclusions on the Pd dispersion and therefore efficiency ist still missing. This information would allow to compare with state of the art homogeneous catalysts for the target reaction. As stated by the authors themselves, results are not sufficent to extract activation energies or other kinetic parameters. I would suggest to complete the data and publish a complete and closed story rather than implying a "to be continued" episode.
Author Response
Response to Reviewer Comments
Reviewer 1:
Point 1: The modifications improved the paper to some extent, however in my opinion this work is still incomplete. The authors included TON and TOF for some samples, however the most interesting relation TON/TOF as function of Pd content, which would allow to draw conclusions on the Pd dispersion and therefore efficiency ist still missing. This information would allow to compare with state of the art homogeneous catalysts for the target reaction.
Response 1: Thank you for your comment. After thorough checking, it was some error in the calculations according to the provided formula in manuscript. We come out with the revised and updated Table 4 along with the related explanation. We compare performance of the catalysts with several reported articles on Pd based heterogeneous catalysts as well as homogeneous catalysts. However, its hard to find Pd complex homogeneous catalysts performance in FA decomposition at similar condition with study in this manuscript.
Point 2: . As stated by the authors themselves, results are not sufficent to extract activation energies or other kinetic parameters. I would suggest to complete the data and publish a complete and closed story rather than implying a "to be continued" episode.
Response 2: Thank you for your concerns and suggestion. We try our best to collect and gather the appropriate data. The Arrhenius plot was provided in the manuscript (In Figure 15(b)) with related explainations (highlighted with yellow color in text).

Reviewer 3 Report
The authors have answered all my concerns, and the manuscipt can be published after checking the reference format carefully.
Author Response
Response to Reviewer Comments:
Reviewer 2:
Point 1: The authors have answered all my concerns, and the manuscipt can be published after checking the reference format carefully.
Response 1: Thank you for the comments. The references format have been carefully checked and revised according to the journal guidelines.

Round 3
Reviewer 1 Report
The additional modifications improved the paper significantly, one thing that should be discussed in my opinion, is the qualitytively different behaviour of the 3% Pd sample. Comparing Figures 12-14, this sample seems to have a larger activation energy than the two other samples. In Line 354 "passivation" was given as explanation for lower activity at high temperatures. This would also explain the comparatively small observed activation energy. However, based on the data shown in Figure 12-14, the 3% Pd sample should have less passivation. Maybe it would be worthwhile to follow up on this feature in the dataset.